# Practices and Awareness Regarding an Infant’s Sleep Environment among Japanese Caregivers: A Cross-Sectional Survey

**DOI:** 10.3390/ijerph21040471

**Published:** 2024-04-12

**Authors:** Ayako Himemiya-Hakucho, Ayumi Taketani, Aoi Nakagawa, Hiroki Sakai, Azumi Shigemoto, Izumi Takase

**Affiliations:** 1Department of Legal Medicine, Yamaguchi University Graduate School of Medicine, Yamaguchi 755-8505, Japan; b004ubv@yamaguchi-u.ac.jp (A.N.); sakai@cetacean.jp (H.S.); jurypico@yamaguchi-u.ac.jp (A.S.); takase@yamaguchi-u.ac.jp (I.T.); 2Undergraduate Courses of Medicine, Faculty of Medicine and Health Sciences, Yamaguchi University, Yamaguchi 755-8505, Japan; i048eb@yamaguchi-u.ac.jp

**Keywords:** awareness, education, knowledge, practice, safe sleep environment, sudden unexpected infant death, suffocation, sudden infant death syndrome

## Abstract

Preventing sudden, unexpected infant death related to sleep, especially suffocation and sudden infant death syndrome, remains challenging globally. To evaluate factors associated with an unsafe sleep environment (SE) for infants in Japan, this cross-sectional study investigated the current status of practices and awareness among caregivers about a safe SE. Two hundred and fifty-four caregivers of infants in Yamaguchi Prefecture participated. Among the caregivers, 96.0% could not thoroughly practice a safe SE, although 65.0% had knowledge about a safe SE. More unsafe SE practices were significantly associated with 8- to 11-month-old infants than with 0- to 3-month-old infants, using the same practice as for an older child than with accessing information or a familiar person than with mass media as the most useful source of information. The differences in having knowledge were not associated with their practice. Many caregivers obtained information about an infant’s SE from mass media and a familiar person. They preferred education via a face-to-face method by medical experts to raise awareness about a safe SE. Thus, efforts need to be developed in Japan in which experts who directly attend to caregivers can truly educate them to ensure that caregivers are continuously aware of the importance of an SE.

## 1. Introduction

Sudden unexpected infant death (SUID), also called “sudden unexpected death in infancy,” is a term used to describe any sudden unexpected death, whether explained or unexplained, that occurs during infancy [1]. SUID includes death due to an external cause such as suffocation, asphyxiation, and abuse; death due to diseases such as infection, metabolic diseases, and arrhythmia-associated cardiac channelopathies; sudden infant death syndrome (SIDS); and death due to an unspecified cause [1]. SIDS is a cause assigned to infant deaths that cannot be explained after a thorough case investigation, including a scene investigation, autopsy, and review of the clinical history [1]. After SIDS, suffocation and asphyxiation are the second most frequent causes of sleep-related SUIDs, and both types of deaths have many similarities in circumstances and risks of occurrence [2,3,4,5,6].

The Back to Sleep campaign, initiated in the UK in 1991 and in the United States in 1994, emphasized lying infants on the back to sleep to prevent SIDS. The rate of SIDS reduced but plateaued as enhanced death scene investigations and child fatality reviews better elucidated the risk of an unsafe sleep environment [1,7]. In 2013, the National Institute of Child Health and Human Development (NICHD; Bethesda, MD, USA) and the American Academy of Pediatrics (Itasca, IL, USA) expanded a campaign called Safe to Sleep^®^ to reduce the risk of sleep-related SUID [8]. It emphasizes lying infants on their backs to sleep, as well as preparing an appropriate sleep environment (SE) for infants, such as using a firm, flat sleep surface, room sharing without bedsharing, avoiding soft bedding and overheating, and keeping soft objects away from an infant’s sleep area [8]. Many countries have begun a similar campaign. However, preventing sleep-related SUID, especially suffocation and SIDS, has remained challenging globally.

In Japan, the Back to Sleep campaign was initiated in 1997, and campaigns to reduce sleep-related SUID were previously promoted by the Ministry of Health, Labor and Welfare (MHLW; Tokyo, Japan) and the Consumer Affairs Agency (Tokyo, Japan); these organizations primarily focused on SIDS and suffocation, respectively [9,10]. The number of SIDS cases decreased from 44.3 per 100,000 live births (LBs) in 1995 to 6.2 per 100,000 LBs in 2018 [11]. The number of deaths due to suffocation decreased from 19.5 per 100,000 LBs in 1995 to 7.9 per 100,000 LBs in 2010 [12]. In recent years, 74 SIDS cases (9.1 per 100,000 LBs) in 2021 [11] and 118 deaths by suffocation during the sleep of infants under 1 year old from 2016 to 2020 [13] have been reported. These reports suggest that the previous efforts have had measurable success. However, the number of diagnoses of death due to an unspecified cause has increased because, since 2005, SIDS has not been diagnosed without an autopsy examination, based on a revision in the diagnostic guidelines by the MHLW [12,14]. Thus, cases of SIDS and suffocation without adequate investigation of the circumstances of deaths are presumedly included in “death due to unspecified cause” in a significant number of cases.

The actual conditions of an SE for infants in Japanese families have been revealed by several surveys conducted in recent years [15,16,17]; a large proportion of respondents used soft and/or loose bedding such as pillows, quilts, blankets, and toys. Kato et al. [16] investigated the SE of 466 infants in three areas from 2017 to 2019 and reported that, among 30% of early-month-old infants, the mother or parents practiced bedsharing. They also reported that more than one-half of parents did not place their infants on their backs when they noticed that the infant was sleeping prone [16]. These reports suggest that the difficulty caregivers have in practicing a safe SE (SSE) for their infants has not been adequately resolved, and implementing an SSE has remained challenging. We believe that understanding whether caregivers are aware of “SSE for infants,” what sources of information they rely on for preparing their infant’s SE, and how they want to be educated about their infant’s SE is important to reduce the number of suffocation and SIDS cases. Surveys focusing on these problems [18,19,20,21,22] and studies [23,24,25] evaluating the preventive interventions for caregivers by medical experts, childcare workers, and others have been reported in several countries. However, no such reports exist in Japan. Hence, the aim of this cross-sectional study was to investigate the current status of practices and awareness about SSE among caregivers to evaluate the factors associated with infants’ unsafe SE in Japan. Based on the findings, we hope to propose future approaches that are needed to reduce SUID in Japan.

## 2. Materials and Methods

### 2.1. Participants and Data Collection

The participants of this study were persons caring for infants under the age of 1 year in Yamaguchi Prefecture, Japan. They were recruited between November 2022 and March 2023. The caregivers accessed the study’s instructions, which were designed using Google Forms (Google, LLC, Mountain View, CA, USA), via a quick response (QR) code on flyers placed in childcare support centers, pediatric clinics, and childcare facilities. Caregivers answered a questionnaire online. This was a convenience sample.

We received responses from 254 caregivers for the analysis. In recent years, the annual number of births in Yamaguchi Prefecture has been approximately 8000 [26]. Thus, this sample size closely corresponded to the probability that each question would be correctly answered by 78% of the respondents when calculated for a target population of 8000, with a 5% margin of error and a 95% confidence interval.

### 2.2. Survey Variables

We conducted an exploratory survey that included questionnaires about current practices and awareness related to an SE when an infant slept at night in the participant’s family. The questionnaire consisted of 25 items used to measure (1) the basic characteristics of the participant and their infant (3 items), (2) the current status of the environment (12 items), and (3) the participant’s level of awareness (10 items) (Appendix A).

The basic characteristics of the participants and their infants included age and the relationship between the caregiver and infant. The current status of the infant’s SE was measured using 12 items, which included room sharing and the corresponding reasons, bedding type (e.g., futon) and the corresponding reasons, use of quilts in the cold season, firmness of bedding (futon) in the cold season, objects in the infant’s sleep area (excluding quilts), and sharing of a surface and/or quilt when using bedding designed for adults. Caregiver awareness of the infant’s SSE was measured by using 10 items, which included access to information about the SE before and after the birth, useful sources of information and the most useful source, safety awareness regarding an infant’s SE, knowledge of the ideal SSE, the most reliable organization to learn about SSE, and the most desirable measures of raising awareness to learn about the SSE. Safety awareness was measured on a 5-point scale from 1 (i.e., “completely unaware”) to 5 (i.e., “strongly aware”). Caregivers’ knowledge of an SSE was measured by whether they had seen and/or heard similar information, such as the “ideal image of an SE in infancy” [27], which was used in this questionnaire and originally quoted from a picture in the Safe to Sleep^®^ campaign.

### 2.3. Statistical Analysis

Data from the survey were screened for completeness, and only a few data were missing. The numbers and percentages contributing to each variable are described within the tables. Comparisons of knowledge about the ideal infant’s SSE with the other categorical variables were examined by using Fisher’s exact test or the chi-squared test.

Multivariable logistic regression models were prepared to estimate each factor associated with unsafe SE practices. The objective variables were participants who selected the “unsafe” categories for more than three of six items (Model 1, *n* = 128) or five of six items (Model 2, *n* = 44) on infant SE, excluding room sharing. The “unsafe” categories in each item were defined as follows: “surface designed for adults” for bedding type; “yes” for the use of quilts in the cold season; “soft” for bedding surface in the cold season; “yes” for objects in the infant’s sleep area; “always,” “often” or “occasionally” for bedsharing; or “yes” for sharing of quilts. Model 1 was designed to assume that the respondents answering “surface designed for infants” in bedding type would select the “unsafe” categories for the remaining three variables. The explanatory variables with no multicollinearity and high contribution to an “unsafe” SE as the objective variable were selected after screening the variables. We followed standard methods to estimate sample size for the analyses; at least 10 outcomes were needed for each included explanatory variable. The adjusted odds ratios (AORs) and 95% confidence intervals (CIs) for four variables in Model 1 and Model 2, respectively, were examined.

All statistical analyses were conducted using JMP Pro for Windows (ver.16; JMP Statistical Discovery LLC., Cary, NC, USA). A *p*-value <0.05 was statistically significant.

### 2.4. Ethics Statement

The participants were informed of the following: their participation in the study was voluntary; the response would be submitted without names—thus, the participants would not be identified; participants could not withdraw from the study after submitting their responses; consent to participate in the study would be considered approval on submitting their responses; they would have no cost for participating in the study; and they would not suffer any disadvantages for not participating in the survey. This study was conducted in accordance with the Declaration of Helsinki and was approved by the Ethics Committee of Yamaguchi University Hospital (Yamaguchi, Japan; approval no. 2022-142).

## 3. Results

### 3.1. Basic Characteristics of the Participants

Sixty (23.6%) participants were aged 20–29 years, 69 (66.5%) participants were aged 30–39 years, and 25 (9.8%) participants were aged 40–49 years. In regard to the relationship with the infant, 240 (94.5%) participants were mothers, and 14 (5.5%) participants were fathers. The median [interquartile range] age of the infants was 6.0 [4.0–8.3] months old. Forty-eight (18.9%) infants were 0–3 months old, 121 (47.6%) infants were 4–7 months old, and 85 (33.5%) infants were 8–11 months old.

### 3.2. The Current Status of Practices for an Infant’s SE

The main results of the responses regarding the current status of practices for an infant’s SE are shown in Table 1. With regard to the sleeping room at night, 92.9% of participants always slept with their infants in the same room. Even among the 10 (3.9%) participants who responded “never,” six participants (five fathers) had another caregiver (e.g., mother) sleep with the infant in the same room, but four infants slept in a separate room from the caregiver or slept in a child’s room with their sibling.

In regard to the bedding type, the proportions of the use of a surface designed for infants and a surface designed for adults were approximately one-half of the participants at 49.8% and 49.4%, respectively. For 0- to 3-month-old infants, a surface designed for infants was used significantly more than a surface designed for adults (66.7% vs. 33.3%, *p* < 0.001). For 8- to 11-month-old infants, a surface designed for infants was used significantly less than a surface designed for adults (34.5% vs. 65.5%, *p* < 0.001). Common reasons for using a surface designed for adults were that caregivers had stopped using the crib they originally used (55.1%), and they did not have problems using the surface designed for adults (44.1%). Other minor responses included the following: no space for a crib, the cost of a crib, co-sleeping, and breastfeeding in the lying down position.

Moreover, the common reasons for stopping the use of a crib were as follows: the infant had difficulty falling asleep in a crib but fell asleep more easily while co-sleeping with the caregiver, and caring for an infant in a crib during bedtime at night became difficult. In addition, some participants responded that the crib rental period was too short.

In the cold season, 90.6% of the participants used quilts for their infants, and 36.6% of them used multiple quilts; however, only 9.4% of them did not use any quilt. A wearable blanket was used by 20 (7.9%) participants. Eight (3.1%) infants were dressed in wearable blankets, as well as covered with quilts. Among the participants, 74.4% of them prepared a firm bedding and 25.6% of them prepared a soft bedding for their infants.

Among all participants, 25.2% had the following objects (excluding quilts) in their infant’s sleeping area: pillows (64.1%), care supplies such as clothing and diapers (43.8%), items for the caregiver or cushions (31.3%), and toys (21.9%).

Among participants who used a surface designed for adults, 89.0% always practiced bedsharing, but only one participant did not practice it. Of these, 53.5% of participants shared quilts designed for adults with their infants.

The proportions of “unsafe” categories selected among the six items on infant’s SE, excluding room sharing, were as follows: zero items for 11 (4.3%) participants, one item for 78 (30.7%) participants, two items for 37 (14.6%) participants, three items for 38 (15.0%) participants, four items for 46 (18.1%) participants, five items for 34 (13.4%) participants, and all items for 10 (3.9%) participants (Table 2).

### 3.3. Awareness about an Infant’s SSE

The results of the responses to awareness about an infant’s SE are shown in Table 3. The number of participants who obtained information on preparing an infant’s SE was 137 (53.9%) before the infant’s birth and 112 (44.1%) after the infant’s birth. For sources of information before an infant’s birth, 24.1% of respondents referred to local or national governments; 27.7% to obstetrics and gynecology (OB/GYN) hospitals; 43.8% to familiar persons; and 86.9% to mass media, including web media. For sources of information after an infant’s birth, 10.7% of respondents referred to childcare facilities; 18.8% to local or national governments; 19.6% to OB/GYN or pediatric hospitals; 40.2% to familiar persons; and 81.3% to mass media including web media. Mass media, including web media, were commonly the most useful information source in both periods (58.8% and 67.0%), respectively. In contrast, 31 (12.2%) participants before the infant’s birth and 52 (20.5%) participants after the infant’s birth did not obtain information to prepare for their infant’s SE. In addition, 86 (33.9%) participants before the infant’s birth and 90 (35.4%) participants after the infant’s birth continued the same caregiving practices as when they had cared for their older children.

In regard to the most reliable organization to learn about an infant’s SSE, 158 (65.2%) participants answered OB/GYN or pediatric hospitals, 44 (17.3%) participants answered local or national governments, 30 (11.8%) participants answered childcare facilities, and 18 (7.1%) participants answered “none.” In regard to the most desirable measures for raising awareness to learn about SSE in infants, 123 (48.4%) participants answered face-to-face verbal explanations, 47 (18.5%) participants answered joining hands-on workshops, 44 (17.3%) participants answered distributing leaflets or flyers, and 40 (15.8%) participants answered watching instructional videos. In regard to safety awareness in an infant’s SE, 94 (37.2%) participants answered 5 (“strongly aware”), 118 (46.6%) participants answered 4 (“aware”), and 5 (2.0%) participants answered 2 (“less aware”).

In regard to knowledge about an infant’s SSE, 165 (65.0%) participants had seen and/or heard similar information as the “ideal image of the SE in infancy” [27]. They had the following characteristics compared with individuals who had never seen and heard this information: a significantly higher proportion was aged 20–29 years, and a significantly lower proportion was aged 40–49 years; the proportion of mothers was significantly higher than that of fathers; a significantly higher proportion of caregivers used a firm bedding surface, and a lower proportion tended to place objects in the infant’s sleep area; a significantly higher proportion obtained information about an infant’s SE before the infant’s birth, whereas the proportion who performed the same caregiving practices as for the older children tended to be lower, and a significantly lower proportion of individuals did not have an organization to depend on to learn about the infant SSE (Table 4). However, no significant difference existed between the groups regarding the most useful source of information about an infant’s SE before and after the infant’s birth (*p* = 0.780 and *p* = 0.182, respectively).

### 3.4. Factors Associated with Unsafe SE Practices for Infants

Significant explanatory factors for unsafe SE practices for infants were identified by using two types of multiple logistic regression models (Table 5). First, in Model 1, practices with more than three unsafe categories were significantly associated with 8- to 11-month-old infants (AOR, 15.38; 95% CI, 3.12–75.88; *p* < 0.001), utilizing a familiar person as the most useful source of information after the infant’s birth (AOR, 4.13; 95% CI, 1.28–13.31; *p* = 0.017), and not having a strong awareness about safety (AOR, 3.75; 95% CI, 1.48–9.49; *p* = 0.005). Second, in Model 2, practices with more than five unsafe categories were significantly associated with the father (AOR 7.96, 95% CI 2.40–26.33, *p* < 0.001), maintaining the same practices as for the older child before the infant’s birth (AOR, 2.68; 95% CI, 1.26–5.69; *p* = 0.010), and not having a strong awareness about safety (AOR, 2.76; 95% CI, 1.21–6.31; *p* = 0.016).

## 4. Discussion

This cross-sectional study was conducted to show the current status of practices and awareness among caregivers about a safe SE to evaluate factors associated with an unsafe SE for infants in Japan. The results indicated that most caregivers could not thoroughly practice infant SSE, and many caregivers had knowledge about an SSE; however, having this knowledge did not always result in an appropriate practice. Furthermore, high rates of unsafe SE practices were significantly associated with caregivers of 8- to 11-month-old infants than caregivers of 0- to 3-month-old infants; with using the same caregiving practices as for an older child than with seeking information; or perceiving a familiar person, rather than with mass media, as the most useful source of information. In addition, differences in having knowledge were not associated with the caregivers’ practice. These findings are discussed in the following sections.

### 4.1. The Problems of Practicing Infant SSE Reported by Caregivers in Japan

Most caregivers in this study did not thoroughly implement infant SSE practices recommended by Safe to Sleep^®^ [8], which was consistent with the results of several similar surveys conducted in Japan in recent years [15,16,17]. In particular, the use of quilts was prevalent in Japan, and many caregivers who prepared a surface designed for adults also shared quilts with their infants. These findings may be because of the fact that the Japanese government’s policy about sleep-related SUID prevention has not been as strict as that of Safe to Sleep^®^. For example, the poster and leaflet produced by the MHLW depict an infant with a light quilt [9]. Moreover, most respondents were mothers, and more acts of unsafe SE were significantly associated with fathers. These findings suggested that mothers are the primary caregivers of their infants, and fathers may have less knowledge about SSE than do mothers. Therefore, fathers in Japan should be more actively involved in the preparation and practices of their own infant’s SE.

Many caregivers answered that they had difficulty putting their infants to sleep and caring for them in cribs, or they answered favorably about using a surface designed for adults. In addition, many older infants were bedded on a surface designed for adults, and 8- to 11-month-old infants also had a higher risk of having an unsafe SE than 0- to 3-month-old infants. These findings suggested that, even if the safety of an infant’s SE is insufficient, the fact that the caregiver is able to interact with an infant every morning may foster the caregiver’s belief that the current SE has no problems for their infant. Furthermore, most caregivers using a surface designed for adults shared the surface, and many caregivers shared quilts with their infants. This finding may be related to the traditional Japanese lifestyle of sleeping together in a room with several futons laid out on the floor in rows without clear boundaries between family members or sleeping to share one or more futons with several family members [28]. Reports indicate that such a cultural and social background has influenced the practice of infant SSE in Japan and in other countries (even in the United States, where Safe to Sleep^®^ has been developed) [29,30,31]. The fact indicates the importance and difficulty of promoting a campaign for sleep-related SUID prevention in relation to a caregiver’s life, background, or circumstances.

### 4.2. Awareness about an Infant’s SSE among Caregivers in Japan

More unsafe SE practices were significantly associated with using the same SE that was used for an older child in this study. A previous study of 259 forensic autopsy cases of SUID in Japan showed a higher odds ratio of its occurring in fourth- and later-born infants [6], which is similar to our findings. In addition, more unsafe SE practices were significantly associated when a familiar person was the most useful source of information about an infant’s SE after the infant’s birth. Support from familiar people basically seems to have many beneficial effects on parenting. However, a survey in the United States reported the prevalence of any advice received from the family or media was 20–56% for nearly all care practices, and the advice given was often inconsistent with NICHD recommendations [32]. Other studies suggest that sharing the experiences of familiar people who have not improved or updated their knowledge could be counterproductive [19,33]. Promoting raising awareness to make individuals who have experienced parenting renew their knowledge and awareness about their own “achievements” is needed.

In this study, the most useful source of information to prepare for an infant’s SE was commonly mass media, which included web media, followed by people familiar with a caregiver. Public institutions, including hospitals, were the lowest in rank. The findings were not associated with whether caregivers had seen or heard of the recommended image. Similar trends were reported in surveys in France [21], Portugal [34], and Malaysia [35]. However, an Australian study [36] demonstrated that 49% of people’s primary source of safe sleep advice was a nurse or midwife, compared to 1.3% from mass media, including social media. Japanese medical professionals should learn from Australian research and be aware of their role in educating caregivers about SSE for infants.

In this study, caregivers who had seen or heard of images of an infant’s SSE, recommended by Safe to Sleep^®^, were more likely to obtain information to prepare an infant’s SE before the birth than were caregivers who had not seen or heard such images. This finding suggested that their willingness to learn can contribute to knowing the appropriate information to some degree. However, our findings showed that having seen and/or heard of the image of an ideal infant’s SSE did not exactly result in the appropriate practice. In addition, the differences in knowledge between mothers and fathers or age exposed the limitations of the practice, depending strongly on individual motivation and awareness for the prevention of sleep-related SUID. Reports [28,37,38,39,40,41] indicate that traditional or cultural practices, exhaustion and fatigue due to parenting, personal circumstances, beliefs, and impractical advice or lack of understanding could be sufficient reasons to commit to unrecommended practices, even if caregivers know the recommended information. This tendency was similar for the caregivers in Japan in our survey.

### 4.3. Promoting Infant’s Safe Sleep Environment Practices among Caregivers in Japan

Togari et al. [42] proposed their recommendations in 2019 for preventing SUID/SIDS, taking into consideration Japanese cultural and social background. Their administrative decision consisted of seven items. Their recommendations did not mention bedding type and surface firmness, and they did not prohibit bedsharing with other family members in a uniform manner [42]. They may have considered previous studies [7,43] that showed the risk of SIDS would not increase if a caregiver simply shared bedding with an infant without other unsafe SE practices or that showed bedsharing with an infant could be beneficial for breastfeeding or improving the quality of a mother’s sleep. The campaign leaflet of the current Japanese government policy recommends the following to prevent SIDS: (1) put the infant on the back until 1 year of age, (2) try to breastfeed as much as possible, and (3) stop smoking. It also recommends the following to prevent suffocation during sleep: (1) put your infant in a crib and keep the fence up; (2) use futons, mattresses, and pillows that are firm and a light quilt; and (3) do not place anything covering the mouth and nose or winding around the neck [8]. In regard to bedsharing, the instruction is limited to a remark about caregivers not putting pressure on their infant with the body or arms [8]. A previous self-administered questionnaire survey of 895 mothers during their infant’s 1-, 4-, or 10-month health checkups in Japan reported that 10.6% of the mothers who practiced co-sleeping had faced infant suffocation incidents, and the frequency of occurrence was significantly more in mothers of 1-month and 4-month-old infants compared with mothers of 10-month-old infants [15]. Previous studies have shown that bedsharing is inappropriate if parents consume alcohol, take drugs, or smoke or if the infant is preterm. They also have shown that sofa-sharing is not a safe alternative to bedsharing [7]. Therefore, in some countries, recommendations do not uniformly prohibit bedsharing with an infant with other family members [35,44]. We believe that providing caregivers with the knowledge associated with bedsharing, in particular, hazardous conditions, is necessary if the benefits of bedsharing are to be emphasized in our country.

This study showed that an infant’s SE may change over months, depending on the circumstances of each caregiver. Some previous studies [32,35,39] have indicated that experts should not take a draconian stance that an inappropriate SE will never be tolerated; they should instead be open to practicing together “relatively safe” or “slightly safer” SE for individual caregivers with their own values and circumstances. This suggestion may be informative in our country in educating caregivers about an infant’s SSE. Moreover, a suggestive divergence was that most respondents obtained information about an infant’s SE from the mass media and familiar person, whereas many of them preferred education via a face-to-face method or by experts in OB/GYN or pediatric hospitals for raising awareness about infant SSE. An interpretation of these findings is that inadequate education of caregivers by an OB/GYN, pediatric hospitals, or the government contributed to their reliance on the mass media or a familiar person. A cross-sectional survey conducted among 137 pregnant women in Japan in 2017 showed that they were poorly aware of and poorly intent on implementing safety practices for preventing sleep-related suffocation [45]. First of all, educating precaregivers by experts during the pregnancy period before they prepare for their infant’s SE is important. Participating in an education program about infant SSE could increase knowledge and intended adherence, but these changes may not be maintained after the baby is born [46]. Therefore, promoting caregivers’ awareness of infant SSE via a continuous and repetitive approach from pregnancy to the parental period can be important. However, interventions that focus solely on giving information are unlikely to produce a meaningful change in a high-risk population. Open conversations tailored to the needs of families and focused on understanding why and when parent(s) do or do not follow safer sleep guidance appear to be a promising method of promoting safer sleep practices [47,48].

We hope that continuous and repetitive awareness-raising programs will be developed in Japan, in which (1) appropriate information is first actively provided and shared with caregivers during pregnancy and (2) discussions on improved measures based on the caregivers’ experiences, circumstances and perspectives, can be provided during parental care. To develop such a program, various organizations involved with children, such as the government, OB/GYN and pediatric hospitals, and childcare facilities cooperating with each other are essential. Safer sleep discussions with families may be best delivered by professionals who have established an ongoing trusting relationship with the parent [48]. However, our previous survey among administrative centers involved in child support, OB/GYN clinics, and pediatric clinics in Yamaguchi Prefecture revealed that the proportion of parental education on SSE was 100%, 54.0%, and 59.2% of the responses, respectively, and the proportion of institutions that had awareness of information such as “ideal image of the SE in infancy” was 58.3%, 29.2%, and 36.7% of responses, respectively [49]. This survey’s findings suggest that medical institutions do not sufficiently recognize the importance of active involvement in raising awareness of SSE and that evidence of SSE is not sufficiently understood by administrators and medical practitioners [49]. In promoting strategies for sleep-related SUID prevention in Japan, the government and medical institutions involved must be aware of other perspectives and recommendations, such as Safe to Sleep^®^, so that they can serve caregivers who may have a wide variety of information.

This study had some limitations. First, we did not conduct a pilot testing measure to assess the feasibility and acceptability of the survey before administering it. In addition, this study was conducted in a single prefecture in Japan. The Yamaguchi Prefecture is a regional area; therefore, responses to this survey may differ from those of the capital cities of Tokyo and other urban areas; the results of this survey cannot be directly attributed to Japan as a whole. To resolve this limitation, studies with data from surveys in multiple regions in Japan should be conducted in the future. Second, this survey focused on the sleep environment and did not investigate infants’ sleep position. However, the practice status of placing an infant on the back to sleep may have contributed important insights to this study. Third, this survey focused on access to information and awareness and did not investigate the background of the caregivers other than age and relationship. However, race, level of education, socioeconomic status, marital status, smoking status, whether a mother breastfeeds, whether caregivers are first-time parents, whether hazardous conditions exist in bedsharing, and other factors may be relevant to the practice of infant SSE. A survey that includes these backgrounds may identify the target population for which more emphasis should be placed on education on sleep-related SUID prevention. In families where the risks of SUID are already higher, interventional approaches may be required that are not just the continuous and repetitive provision of knowledge [50].

## 5. Conclusions

In this survey, the current status of infant SE practices and awareness related to infant SSE among caregivers in Japan was surveyed, and their associations were analyzed. Our findings revealed that having safety awareness and knowledge about infant SE did not always result in an appropriate practice. In addition, the turning point in an infant’s SE occurred when caregivers did not use a surface designed for infants or changed to a surface designed for adults during infancy. Since 2023 in Japan, the Child and Family Affairs Agency (CFAA) has become primarily involved in the effort regarding children. In the future, the CFAA should establish a system to conduct an adequate investigation of the circumstances of all deaths of sleep-related SUID and should disseminate and share adequate knowledge among medical and health experts or institutions. The experts who have opportunities to directly support caregivers need to educate them so that caregivers will have knowledge about an infant’s SSE and maintain an SSE for their infant. In addition, discussions focused on understanding why and when caregivers do or do not follow safer sleep guidance, based on the caregivers’ experiences, circumstances, and perspectives, are needed to promote safer sleep practices.

## Figures and Tables

**Table 1 ijerph-21-00471-t001:** The current status of practices for an infant’s safe environment.

Variable (*n*)	Category	*n* (%)
The infant and the participant sleep in the same room (254)	Always	236 (92.9)
	Often	7 (2.8)
	Occasionally	1 (0.4)
	Never	10 (3.9)
Bedding type (253)	Surface designed for infants	126 (49.8)
	Surface designed for adults	125 (49.4)
	Unfixed	2 (0.8)
Use of quilts in the cold season (254)	Yes	230 (90.6)
	No	24 (9.4)
Bedding surface in the cold season (254)	Firm	189 (74.4)
	Soft	65 (25.6)
Objects in the infant’s sleep area (excluding quilts) (254)	Yes	64 (25.2)
	No	190 (74.8)
Bedsharing (127) ^†^	Always	113 (89.0)
	Often	11 (8.7)
	Occasionally	2 (1.6)
	Seldom	1 (0.8)
Sharing of quilts (127) ^†^	Yes	68 (53.5)
	No	59 (46.5)

^†^ Questions were answered by participants who did not select “surface designed for infants” for the bedding type.

**Table 2 ijerph-21-00471-t002:** The proportions of “unsafe” categories selected from among six items in the infant’s sleep environment.

Number of the Items	*n* (%)
0	11 (4.3)
1	78 (30.7)
2	37 (14.6)
3	38 (15.0)
4	46 (18.1)
5	34 (13.4)
6	10 (3.9)

**Table 3 ijerph-21-00471-t003:** Awareness about the infant’s safe sleep environment.

Variable (*n*)	Category	*n* (%)
Access to information about the infant’s SE		
Before the infant’s birth (254)	Yes	137 (53.9)
	No	31 (12.2)
	Used the same caregiving practices as for an older child	86 (33.9)
After the infant’s birth (254)	Yes	112 (44.1)
	No	52 (20.5)
	Used the same caregiving practices as for an older child	90 (35.4)
Most useful source of information about the infant’s SE ^†^		
Before the infant’s birth (136)	Local or national government	10 (7.3)
	OB/GYN hospital	16 (11.8)
	Familiar person	30 (22.1)
	Mass media, including web media	80 (58.8)
After the infant’s birth (109)	Childcare facility	3 (2.7)
	Local or national government	4 (3.7)
	OB/GYN or pediatric hospital	8 (7.3)
	Familiar person	21 (19.3)
	Mass media, including web media	73 (67.0)
Most reliable organization to learn about an infant’s SSE (254)	OB/GYN or pediatric hospital	158 (65.2)
	Local or national government	44 (17.3)
	Childcare facility	30 (11.8)
	None	18 (7.1)
	Others	4 (1.6)
Most desirable measures of raising awareness to learn about the infant’s SSE (254)	Face-to-face verbal instruction	123 (48.4)
	Joining hands-on workshops	47 (18.5)
	Distributing leaflets or flyers	44 (17.3)
	Watching instructional video	40 (15.8)
Safety awareness (253)	5 (“Strongly aware”)	94 (37.2)
	4 (“Aware”)	118 (46.6)
	3 (“Neither aware nor unaware”)	36 (14.2)
	2 (“Less aware”)	5 (2.0)
	1 (“Completely unaware”)	0 (0)
Seen and/or heard about the ideal SSE for infants (254)	Yes	165 (65.0)
	No	89 (35.0)

SE, sleep environment; SSE, safe sleep environment; OB/GYN, obstetrics and gynecology. ^†^ Questions were answered by the participants who selected “yes” in “access to information”.

**Table 4 ijerph-21-00471-t004:** Comparison of the variables with knowledge of the ideal safe sleep environment for infants.

Variable	Category	Have Seen or Heard	Have Never Seen or Heard	*p*
Age of the participants (y)	20–29	47 (28.5)	13 (14.6)	<0.001
	30–39	111 (67.3)	58 (65.2)	
	40–49	7 (4.2)	18 (20.2)	
Relationship with the infant	Mother	160 (97.0)	80 (89.9)	0.018
	Father	5 (3.0)	9 (10.1)	
Surface of bedding in the cold season	Firm	132 (80.0)	57 (64.0)	0.005
	Soft	33 (22.0)	32 (36.0)	
Objects in the infant’s sleep area (excluding quilts)	Yes	36 (21.8)	28 (31.5)	0.098
	No	129 (78.2)	61 (68.5)	
Access to information about the infant’s SE before the infant’s birth	Yes	99 (60.0)	38 (42.7)	0.017
	No	20 (12.1)	11 (12.4)	
	Used the same caregiving practices as for the older child	46 (27.9)	40 (44.9)	
Access to information about the infant’s SE after the infant’s birth	Yes	82 (49.7)	30 (33.7)	0.044
	No	29 (17.6)	23 (25.8)	
	Used the same caregiving practices as for the older child	54 (32.7)	36 (40.5)	
Most reliable organization to learn about the infant’s SSE	OB/GYN or pediatric hospital	108 (65.5)	50 (56.2)	0.069
	Local or national government	31 (18.8)	13 (14.6)	
	Childcare facility	16 (9.7)	14 (15.7)	
	None	7 (4.2)	11 (12.4)	
	Other	3 (1.8)	1 (1.1)	

The data are presented as the *n* (%). SE, sleep environment; SSE, safe sleep environment; OB/GYN, obstetrics and gynecology.

**Table 5 ijerph-21-00471-t005:** Multivariable logistic regression analysis of factors associated with unsafe sleep environment practices.

Variable	Category	Model 1			Model 2		
		AOR	95% CI	*p*	AOR	95% CI	*p*
Relationship with the infant	Mother (Reference)	-			1		
	Father	-			7.96	2.40–26.33	< 0.001
Age of the infant (mo.)	0–3 (Reference)	1			-		
	4–7	3.46	0.80–15.04	0.098	-		
	8–11	15.38	3.12–75.88	< 0.001	-		
Access to information about the infant’s SE before the infant’s birth	Yes (Reference)	-			1		
	No	-			1.40	0.45–4.28	0.560
	Used the same caregiving practices as for the older child	-			2.68	1.26–5.69	0.010
Access to information about the infant’s SE after the infant’s birth	Yes (Reference)	1			-		
	No	1.55	0.16–14.59	0.701	-		
	Used the same caregiving practices as for the older child	2.65	0.10–73.30	0.566	-		
Most useful source of information about the infant’s SE after the infant’s birth	Mass media, including web media (Reference)	1			-		
	Familiar person	4.13	1.28–13.31	0.017	-		
	Government/Hospital/Childcare facility	3.06	0.74–12.61	0.121	-		
Safety awareness about the infant’s SE	5 (Reference)	1			1		
	1–4	3.75	1.48–9.49	0.005	2.76	1.21–6.31	0.016

SE, sleep environment; SSE, safe sleep environment. The objective variable is the participants who selected the “unsafe” categories for three of six items (Model 1, *n* = 128) or five of six items (Model 2, *n* = 44) on an infant’s SSE, excluding room sharing.

## Data Availability

The datasets generated for this study are available on request from the corresponding author.

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
