# Peer review of "Practices and Awareness Regarding an Infant’s Sleep Environment among Japanese Caregivers: A Cross-Sectional Survey"

_ijerph, 2024, doi:10.3390/ijerph21040471_

Round 1

Reviewer 1 Report

Comments and Suggestions for Authors

Overall, this is a well organized first draft. Below are my feedback:

Add in additional goals related to informing directions for future approaches

Specify that this was a convenience sample in your methods

Also in your methods, add in more content about the exploratory nature of this survey and include some of the key content areas covered in the survey

Consider including questions asked or focus areas of questions in a table that complements the written text of the methods section

Add in any content about pilot testing measure to assess feasibility and acceptability of it prior to administration in the methods – if it was not conducted, then add this as a limitation in your discussion section

In the beginning of your discussion section, my suggestion is to present a summary paragraph on the key takeaways from your findings – the consistencies and inconsistencies in findings, summary statements on which findings were and were not statistically significant, which ones were supported / not supported by your hypotheses, and your assessment on the gap / missing link from your study to inform next directions

Was there any content pertaining to infant safe sleep guidelines covered in your study as a point of knowledge / awareness among participants? If not, could add a recommendation to account for awareness of them in combination with campaigns as tools to guide infant safe sleep practices

Share your thoughts in the discussion section on any additional considerations for educational interventions that could involve heightening knowledge and awareness via campaigns, utilization of guidelines, additional health promotion endeavors, etc

Comments on the Quality of English Language

With regard to could be changed to in regard to 

add in age range of 20s rather than in their 20s, same applies for any other age ranges (e.g. 40s)

Reviewer 2 Report

Comments and Suggestions for Authors

The paper describes the findings from a cross-sectional survey of caregivers in Japan, related to their infant sleep practices and knowledge of safe sleep messages. There are some things that need clarification or correction:

Line 30-31: Sudden unexpected infant death (SUID), also called “sudden unexpected death in infancy,” occurs during sleep. This is incorrect. SUID includes deaths which are explained and unexplained. A large proportion of the unexplained deaths under the age of 1 occur during sleep. The authors need to add in the correct definitions of both sudden unexpected death in infancy and sudden infant death syndrome. 

The point the authors make in the introduction about the difficulties with ascertaining an unexplained infant death rate in Japan is important and could form part of the discussion, as monitoring rates from any intervention will be impossible with things as they currently stand. 

The authors rightly bring up cultural practices and surface-sharing but I think they could have been clearer with 1) exactly what is recommended to Japanese families, by whom, and at what time, and 2) how they defined safe and unsafe sleep practices. It is not clear in the paper whether any or all surface sharing was considered as "unsafe". 

The authors need to add in to the limitations section the factors that we know make surface sharing more and less risky and consider how not having any data on these factors has influenced the findings from the survey: 10.1371/journal.pone.0107799 

Line 90-94: Please provide a copy of the survey translated into English. 

There appeared to be no measures of socioeconomic position, please explain why this was not included, when we know that unexplained infant death shows marked inequalities across every other country that has measured it? 10.1093/qjmed/hcac093 

The tables are very difficult to decipher and need lines separating each row. 

Line 168: crib rental period - is this normal in Japan? Can you include information on this in the discussion and whether increasing minimum crib rental periods might help increase safety?

Line 181-185: Please include a table detailing each 'unsafe' item and these proportions for the sample. 

Line 252-253: This seems to say that the Japanese government advice for reducing the risks of SIDS is different to the 'Safe to Sleep' advice, which is from the USA. Is that correct? If so, why did the authors compare practices in Japan with advice from a completely different country?

Lines 346-347: The authors focus solely on advice giving, in a contiuous and repetetive manner. We have fairly good evidence from other countries that this approach might not work very well for the families where the risks are already higher. See https://bmjpaedsopen.bmj.com/content/5/1/e000983 

I wonder if the authors could propose some other more appropriate ways to support families with following safer sleep advice, given their survey findings? 

Why did you exclude room sharing in the regression analysis?

Comments on the Quality of English Language

The English is quite difficult to follow in some areas, just for example: "In addition, 86 (33.9%) participants before the infant’s birth and 90 (35.4%) participants after the infant’s birth did the same as when they had nursed their older children." Nursed in America means 'breastfed', and in the UK it tends to mean 'looked after when ill'.

Reviewer 3 Report

Comments and Suggestions for Authors

Thank you for the opportunity to review  this interesting paper.  My main comment is that there is no mention of sleep position for the infant.  Sleep position is a major risk factor for SUDI – infants sleeping prone or on their side have a much greater risk.  If there are no data on this in the survey the reasons for this must be explained as it’s a significant limitation. 

I cannot comment on the statistical analysis – this paper needs an appropriate review by a statistics editor.  I do not think however that the logistic regression contributes significantly to the paper, the findings are interesting enough without this.

Abstract

Line 21 onwards – this section is not clear and needs rephrasing.

Introduction

Line 33.  SIDS needs to be defined – the current definition is 2004 Krous et al.  It is not a disease,  it is an unspecified cause. This needs to be clarified.

Line 41.  One of the most important safe sleep recommendations is infants lying on their back to sleep.  This must be included in the introduction.

Line 43. Safe sleep campaigns did not originate in the USA – they started in UK and New Zealand, USA campaigns were a few years later. Europe, North America, Australia and New Zealand have similar safe sleep recommendations. This should be corrected.

Line 46.  This paragraph is difficult to follow and needs rephrasing.  A key point is that without post-mortem investigation it is very difficult to determine SIDS rates as these will be highly inaccurate – see Taylor, Garstang et al.  It might be better to show changes in total post-neonatal mortality in addition.

Line 63.  Could you give more detail on the findings from individual surveys of Japanese safe sleep practices? Are there national recommendations for infant sleep in Japan?  Please explain these – or the lack of them.

Line 66. What is a comforter?  This has several different meanings eg favourite toy, pacifier (dummy).  Please explain what you mean by this.

Methods

Line 84.  I didn’t quite understand the description of the sample size calculation but I am assuming there will be a statistical review of this paper.

Line 101. Did the survey include whether the infant was put to sleep on their back (supine)?  This is a very important safe sleep factor, and if it was not included this is a significant weakness of the study.

Line 102. Please explain what a comforter is

It would be helpful to see a table with a summary of the questions asked of parents

Results

Table 1 and 2 the lack of lines mean it is difficult to read which category of response relates to which variable.

Line 169 – comforters?  Are these blankets?  This is confusing.

Line 173 – how did you define soft or firm bedding in your survey?

Discussion

This should include some reference to the literature on parental decision making about safe sleep environments.  Giving parents information alone is not sufficient for many parents.  See recent work by Anna Pease or Trina Salm Ward.

I think you could also be clearer about how you could make safe sleep recommendations for Japan – advising again all co-sleeping (as per USA) is not going to work, as this is a culturally normal practice.  The advice needs to be about risk minimisation when co-sleeping.

Comments on the Quality of English Language

The paper would benefit from some minor language editing, particularly of the abstract.
